# Ratiometric quorum sensing governs the trade-off between bacterial vertical and horizontal antibiotic resistance propagation

**Alvaro Banderas**[1,2¤a¤b]*, **Arthur Carcano**[3,4], **Elisa Sia**[1,2], **Shuang Li**[1,2], **Ariel B. Lindner**[1,2]*

**1** INSERM U1284, Systems engineering and evolution dynamics, Paris, France, **2** Center for Research and Interdisciplinarity, Université de Paris, Paris, France, **3** INRIA Saclay–Ile-de-France, Palaiseau, France, **4** USR 3756 IP CNRS, Institut Pasteur, Paris, France

¤a Current address: Physico Chimie Curie UMR168 CNRS, Institut Curie, Paris, France
¤b Current address: Physique du vivant, Laboratoire Matière et Systèmes Complexes, UMR7057 CNRS, Paris, France
* alvaro.banderas@curie.fr (AB); ariel.lindner@inserm.fr (ABL)

**Data Availability Statement:** All relevant data are within the paper and its Supporting Information files. Python code with mathematical model and

## Abstract

Plasmid-mediated horizontal gene transfer of antibiotic resistance and virulence in pathogenic bacteria underlies a major public health issue. Understanding how, in the absence of antibiotic-mediated selection, plasmid-bearing cells avoid being outnumbered by plasmid-free cells is key to developing counterstrategies. Here, we quantified the induction of the plasmidial sex pheromone pathway of *Enterococcus faecalis* to show that the integration of the stimulatory (mate-sensing) and inhibitory (self-sensing) signaling modules from the pCF10 conjugative plasmid provides a precise measure of the recipient-to-donor ratio, agnostic to variations in population size. Such ratiometric control of conjugation favors vertical plasmid transfer under low mating likelihood and allows activation of conjugation functions only under high mating likelihood. We further show that this strategy constitutes a cost-effective investment into mating effort because overstimulation produces unproductive self-aggregation and growth rate reduction. A mathematical model suggests that ratiometric control of conjugation increases plasmid fitness and predicts a robust long-term, stable coexistence of donors and recipients. Our results demonstrate how population-level parameters can control transfer of antibiotic resistance in bacteria, opening the door for biotic control strategies.

## Introduction

Horizontal gene transfer mediated by conjugative plasmids and integrative conjugative elements (ICEs) is a major cause of the rapid spread of bacterial antibiotic resistance [1]. The process starts with a "mating" stage, which depends on contact through sexual pili or cell–cell aggregation proteins followed by a type IV secretion system [2]. Importantly, expression of these conjugation determinants bears a significant fitness cost, reducing the host's growth rate,

simulations is available at https://gitlab.inria.fr/InBio/Public/efaecalis-ratio.

**Funding:** LabEx -Who am I?- Postdoctoral fellowship to AB (ANR-11-LABX-0071 SEXASYM) (https://www.enseignementsup-recherche.gouv.fr/cid51355/laboratoires-d-excellence.html) Permanent support. Fondation Bettencourt Schueller (https://www.fondationbs.org/) to ABL The funders had no role in study design, data collection and analysis, decision to publish, or preparation of the manuscript.

**Competing interests:** The authors have declared that no competing interests exist.

**Abbreviations:** ccfA, cCF10 pheromone gene; CFU, colony-forming unit; GFP, green fluorescent protein; LTA, lipoteichoic acid; ICE, integrative conjugative element; ODE, ordinary differential equation; Opp, oligopeptide permease; ORF, open reading frame; Prg, pheromone responding gene; RBS, ribosome binding site; Sec10, surface exclusion from pCF10.

implying the existence of a trade-off between the horizontal and vertical modes of plasmid transfer [3]. Indeed, across gram-negative and gram-positive bacteria, plasmid donor populations are generally in a constitutive "off-state," and only after induction by environmental factors or signaling molecules is conjugation activated [4]. Also, a general feature of these systems is that only a few members of the population activate the response [4]. This property suggests that antibiotic-resistant plasmids and ICEs naturally maintain a high proportion of vertical rather than horizontal transfer. On an evolutionary timescale, under constant low availability of plasmid recipients, plasmid variants with lower conjugation rates are expected to gain a fitness advantage by increasing their vertical inheritance through host proliferation. Contrary to that, evolution is expected to favor plasmid variants with increased conjugation rates under conditions of constant high recipient availability [5]. Therefore, the optimal effort that a plasmid invests on horizontal spread depends principally on the social environment under which conjugation control evolved. However, the primary population parameters determining the potential donor–recipient encounter rates, i.e., the densities of donors and recipients, can vary dynamically at faster timescales through growth, dilution, and migration processes. Intuitively, then, plasmid variants able to dynamically monitor mating likelihood and regulate conjugation effort accordingly could evolve.

Antibiotic-resistant plasmids from *Enterococcus faecalis* are a major threat to public health [6] because of the efficiency of their pheromone-sensitive conjugation systems [7] and the multiplicity of resistances they can transfer, including those against the last-line antibiotic vancomycin [8]. Plasmids from *E. faecalis* exhibit the capacity to sense recipient densities [9]. In the pCF10 plasmid and its family members, this mate sensing is achieved with unidirectional signaling based on conserved peptide import–export systems [10–13]. This system is also no exception in terms of the cellular cost of conjugation, given the strong protein expression up-regulation that ensues. Indeed, efficient cell–cell aggregation in this system relies on the high abundance of the aggregation substance from pCF10 (Asc10) protein [14,15], as well as on the expression of >20 pheromone-regulated genes, including the ATP-dependent type IV conjugation machinery [16,17]. Donor populations exposed to a given level of inducer also have the capacity to sort into responding and nonresponding cells—in this case, through a pheromone-dependent bistable switch at the transcriptional level [18].

From an evolutionary perspective, sensing the presence of mates through selection for specificity (Fig 1A, compare left and center panels) is straightforward to interpret because it avoids unproductive donor–donor interactions. Intriguingly, however, in the *E. faecalis* system, two antagonistic plasmid-encoded, pheromone-sensing systems control conjugation. These integrate information about the presence of potential plasmid recipients (mate-sensing) and about plasmid donors (self-sensing) (Fig 1A, right; [16]). Such integration is antagonistic, with recipient-produced cCF10 and donor-produced iCF10 pheromones causing activation and repression of conjugation functions, respectively (Fig 1A, right). Such repressive self-sensing could effectively prevent self-aggregation and unproductive homophilic interactions at high donor densities, in which donor-produced leaky cCF10 (Fig 1A) could accumulate to significant levels. Self-sensing is mediated by basal production of iCF10 during the growth of uninduced donors. This pheromone is essential for keeping the conjugation pathway inactive in donor populations, even if leaky cCF10 accumulates due to growth, as shown by saturated constitutive pathway expression in plasmids carrying a nonfunctional version of iCF10 [19]. We rationalized that the combination of self-sensing and mate sensing could serve another function—namely, conferring to donor cells the capacity to perceive mate availability (ratio sensing) rather than mate concentration only (quorum sensing, [20]). This could enable cells to respond specifically to the population composition (Fig 1B). Intuitively, growth in liquid may increase the rate of accumulation of both cCF10 and iCF10 (Fig 1B, right), but the effects

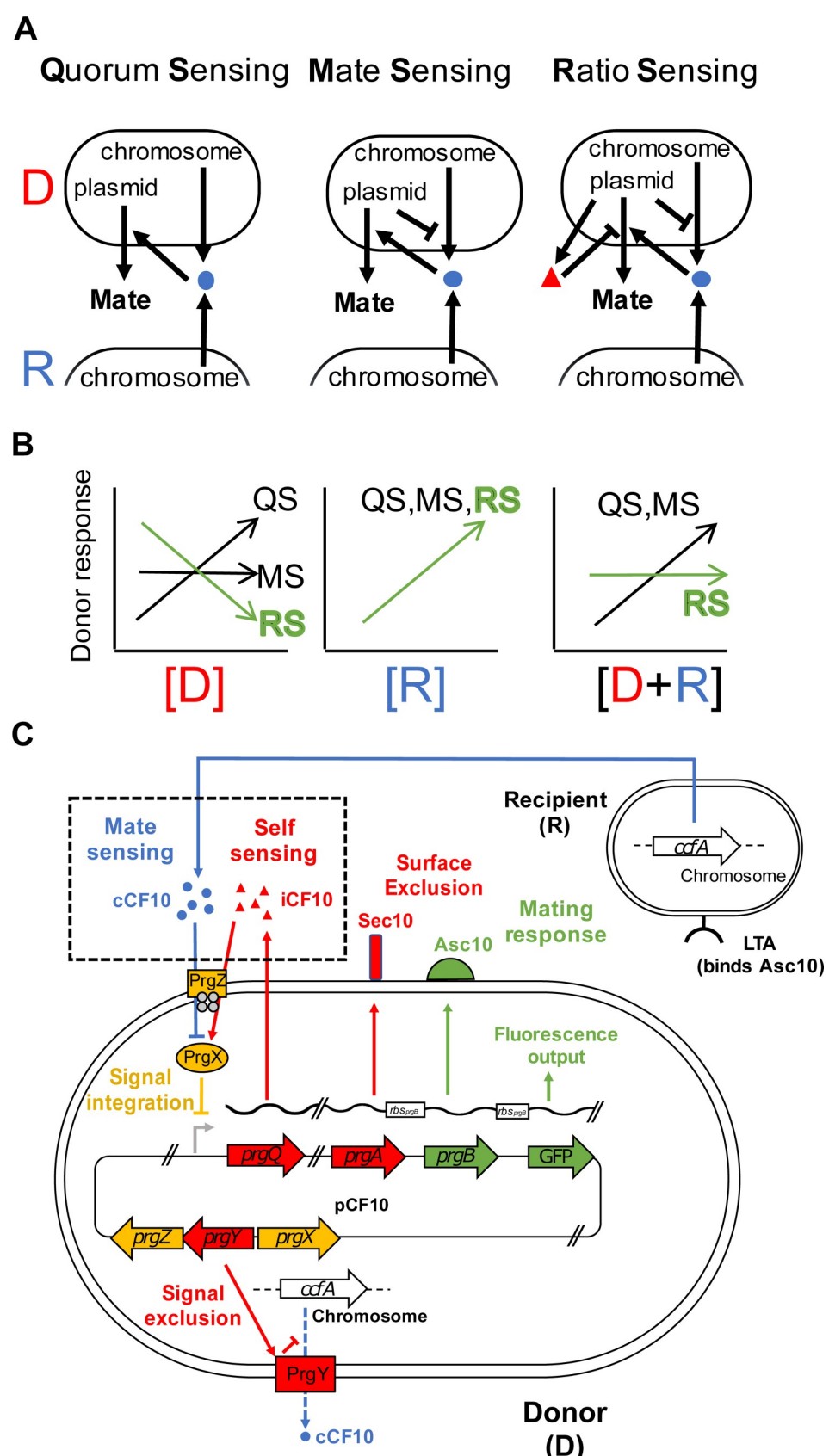

**Fig 1. Population parameter sensing in *E. faecalis* plasmid donors.** (A) Possible topological structures of pheromone sensing in pCF10. Plasmids that perceive cCF10 pheromone (blue balls) from either donors or recipients (left) and consequently activate the pathway in proportion to the total population density (QS) are less efficient at mating than variants that minimize the production of endogenous cCF10 (center, MS). The present-day topological structure involves the presence of an antagonistic extracellular pheromone (iCF10, right). Pheromones cCF10 (blue balls) and iCF10 (red triangles) accumulate in proportion to recipients and donors, respectively. (B) The population parameter disentanglement capabilities of pCF10 might provide a function of iCF10; [D] (left), [R] (center), and [R+D] (right) are sensed differently by each one of the signaling schemes. Contrary to QS and MS, only the RS (green) scheme can distinguish specifically meaningful changes in recipient availability from simple fluctuations in crowding. (C) The mating system of *E. faecalis*. The pCF10 plasmid in donor cells encodes "mating" (preconjugative) functions involved in self-incompatibility (red), which allows avoidance of nonproductive donor–donor interactions in at least three ways. First, by Sec10 (encoded by *prgA*) activity, which minimizes interactions of the neighboring cell-wall–associated aggregation substance (Asc10, coded by the *prgB* gene) with LTA in the cell walls of other donors (not shown) by steric hindrance [21] and keeps those interactions specific for the LTA in recipients (shown). Second, by *prgY*, which restricts production of the cCF10 pheromone (blue) [22], a secreted product of the normal processing of a protein encoded by the *ccfA* gene, encoded in the genome which serves as the main cue used for activation. Finally, by secreting the iCF10 pheromone, which antagonizes the effect of cCF10 at the signal integration level (yellow) through competitive binding to the PrgX transcription factor. PrgZ is responsible for pheromone binding along with internalization by the native Opp system (gray) [10,11]. In this study, the pathway's response was quantified by measuring Asc10-dependent phenotypes, such as adherence to surfaces and sexual aggregate formation, and by monitoring the expression of a GFP reporter cotranscribed with the *prgB* gene [23]. The reporter's RBS (white box on transcript) is identical to that of *prgB*. Functions further downstream of *prgB* (including the conjugation machinery) are not shown. *ccfA*, cCF10 pheromone gene; [D], donor concentration; GFP, green fluorescent protein; LTA, lipoteichoic acid; MS, mate sensing; Opp, oligopeptide permease; Prg, pheromone responding gene; [R], recipient concentration; RBS, ribosome binding site; [R+D], total population concentration; RS, ratio sensing; QS, quorum sensing; Sec10, surface exclusion from pCF10.

produced by each could balance each other out, producing a response that remains insensitive to fluctuations in the degree of cellular crowding.

Here, we demonstrate that density-robust ratiometric control over horizontal plasmid transfer allows antibiotic-resistant *Enterococcus* plasmid donors to estimate the conjugation likelihood in a cost-effective manner, maximizing their fitness. We further suggest that this mechanism robustly stabilizes the population composition in the long term in the face of variation in resource availability.

## Results

### Ratiometric sensing of population composition

To test whether cells are indeed capable of distinguishing population ratio from recipient density (Fig 1B), we quantified pheromone-mediated induction of the pCF10 plasmid during sexual aggregation by performing donor–recipient coincubation experiments and measuring the gene expression response of the pCF10 mating–pheromone pathway using flow cytometry (Fig 1C, S1 Fig) in plasmid donors. For this, we used OG1RF(pCF10-GFP) donors [23]—a bright green fluorescent protein (GFP) reporter strain of pheromone induction (S2 Fig)—to both distinguish fluorescent donors from autofluorescent recipients and to quantify the pheromone responsive gene (prg) *prgB* (encoding Asc10) expression in flow cytometry experiments (S1 Fig). We resolved the dependency of the response on population parameters (composition and total density) across a wide range of values (Fig 2A and 2B). We observed that ratio but not density causes activation of conjugation. This analysis showed that donor cells commit to strong Asc10 expression only when populations are recipient biased, and conversely, they suppress the response in donor-biased populations. Crucially, the fraction of donor cells that show detectable response levels (Fig 2C) correlates with the result of mating assays (Fig 2D) in the wild-type strain in which on the one hand, the mating efficiency reaches 100% (all donors mate once) at highly recipient-biased ratios and, on the other, remains constant within a

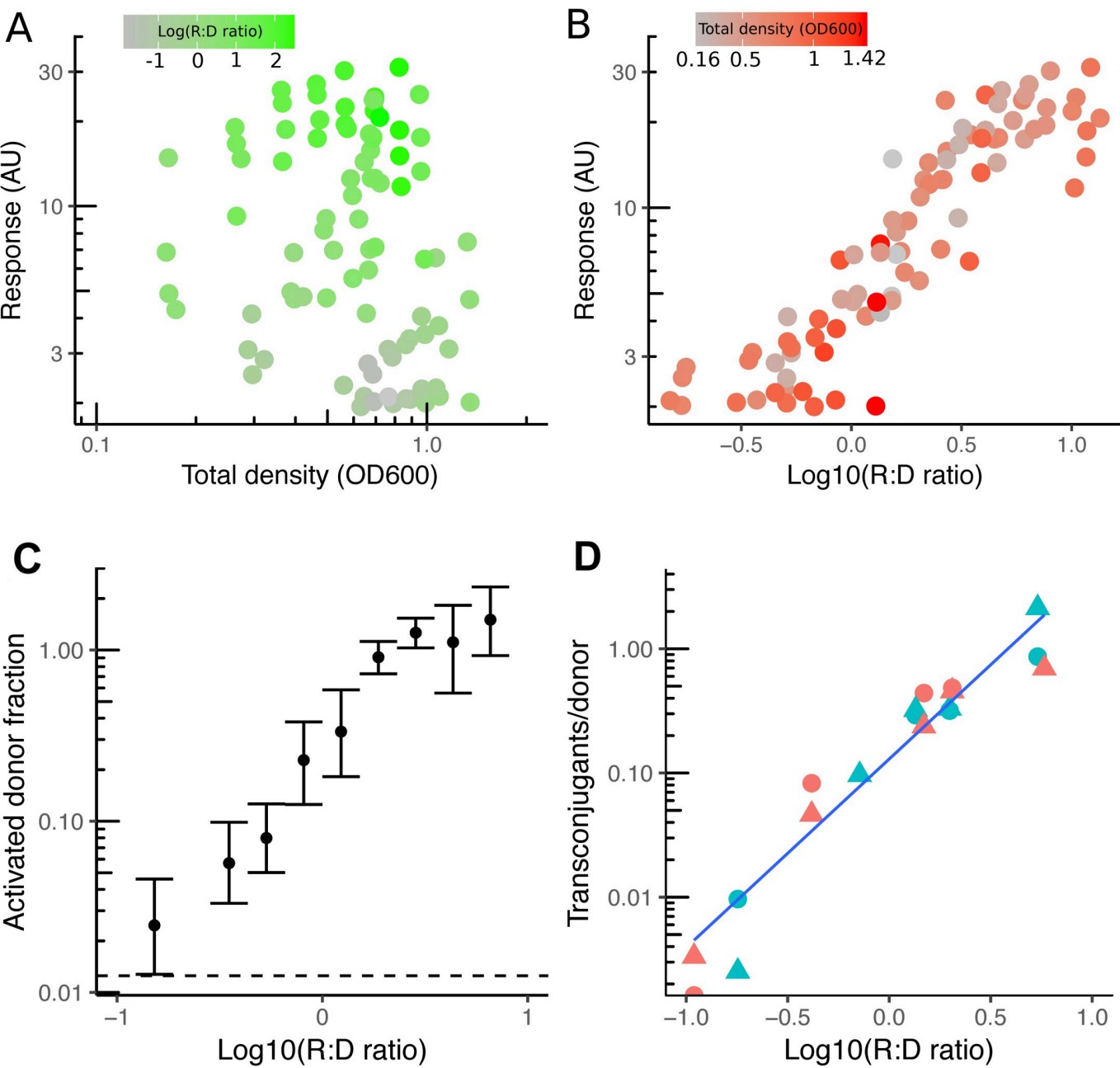

**Fig 2. Population density robust ratiometric sensing in *E. faecalis*.** (A, B) Response (mean GFP fluorescence intensity per cell) of plasmid pCF10 in populations with varying concentrations of emitter and donor cells at 2 hours after mixing and measured by flow cytometry (see *Flow cytometry*). The response of OG1RF(pCF10-GFP) donors is plotted as a function of the two essential population parameters, the [R+D] (A) and R:D ratio (B), each colored according to their counterpart parameter: green for the R:D ratio levels (A) and red for the total density levels (B) (data set shows seven independent experiments). (C, D) The fraction of early (1 hour) activated donors as a function of the R:D ratio from the data set in (A) and (B). Data were regularly binned (10), and the SEM was calculated (error bars). The dashed line is the basal value at R:D ratio = 0 (C). Mating efficiency (see *Mating experiment*) as a function of the R:D ratio at 2 (circles) and 3 (triangles) hours with total densities (OD600) of 0.1 (red) and 0.5 (blue). The underlying numerical data are shown in S1, S2 and S3 Data. GFP, green fluorescent protein; R:D, recipient-to-donor; [R+D], total population concentration; OD, optical density.

5-fold difference in the total population cell density, and sensitive within two orders of magnitude of population composition values.

Our results show that the fraction of donors that mate increases at recipient-biased ratios but remains roughly constant when the total density is varied. This result implies that when

populations are donor biased, the activation of the system occurs in only a subset of cells during aggregation, consistent with the bistable model of pheromone induction [18], and further shows that the donor population commits to conjugation only when in the minority and not at any specific recipient density, i.e., partner sensing is ratiometric and not densitometric.

## Asc10 expression reduces vertical transfer of conjugative plasmids

Ratiometric sensing is reminiscent of the mating system of *S. cerevisiae*, in which the population's sex ratio allows cells to sense the degree of mate competition in the population [24]. In yeast, A-type cells use sex ratio sensing to balance a trade-off existing between commitment to mating and clonal haploid proliferation. However, whether ratiometric sensing relates to the existence of a similar trade-off in *E. faecalis* is unclear. To test whether a trade-off between investment in conjugation and vertical plasmid proliferation exists in *E. faecalis*, we first estimated the initial pheromone concentrations that correspond to the stimulation range observed in the coincubation experiments. Shaken cultures of donors exposed to sufficient cCF10 concentrations tend to self-aggregate, i.e., [cCF10] > 10 nM (S3 Fig). Although a reduction in growth below 10 nM is detectable by optical density measurements (S4 Fig), growth is difficult to measure precisely under shaking conditions, either with optical density or colony-forming unit (CFU) counts, as microaggregates are difficult to disperse. For this reason, we developed an assay to quantify growth in static cultures, in which self-aggregation is not aided by orbital shaking and the pheromone response is detected as Asc10-dependent surface attachment at cCF10 concentrations below pathway saturation (Fig 3A). By quantifying growth under such conditions, we show that much of the growth impairment occurs at pheromone concentrations marking the transition point to surface adhesion and that such impairment increases further only at high, nonphysiological pheromone concentrations (Fig 3B). At concentrations in the micromolar range, we observed a time-dependent decrease in the optical density at the highest cCF10 concentrations, consistent with pheromone-induced toxicity [25]. Importantly, at concentrations lower than approximately 1 pM, cells accumulate as an easily visible dense precipitate that can be easily resuspended, unlike Asc10-expressing cells, which distributed homogeneously as a film in the bottom and remained adhered after resuspension (Fig 3A and 3B). This allowed us to determine that the physiological range of sensing, defined as the pheromone concentrations within which cells can activate Asc10 and avoid self-aggregation (S3 Fig), is roughly two orders of magnitude, consistent with the dynamic range of sensing on the recipient-to-donor ratio scale in coincubation experiments (Fig 2B).

Our results show that a trade-off between vertical and horizontal plasmid transfer exists in *E. faecalis*. This suggests that, similar to budding yeast [24], specifically controlling Asc10 expression according to the recipient-to-donor ratio (a proxy for the mating likelihood) could be selected by evolution. Such pheromone-induced fitness loss is likely due to mild pheromone-induced prgB-dependent toxicity, similar to previous observations in *prgU* knockouts [25]. Using our assay, we show that wild-type pCF10 exhibits an approximately 20% maximal growth reduction, within the sensitive range of the pathway (S5 Fig), whereas a *prgU* knockout is more severe at higher pheromone concentrations (S6 Fig). Balancing the observed trade-off to maximize fitness might therefore be the primary role of ratio sensing.

## Ratio sensing can increase plasmid fitness and facilitates donor–recipient coexistence

To understand the role that ratiometric regulation of conjugation has on plasmid fitness, we built a simplified dynamic model consisting of donor and recipient populations described by two coupled ordinary differential equations (ODEs).

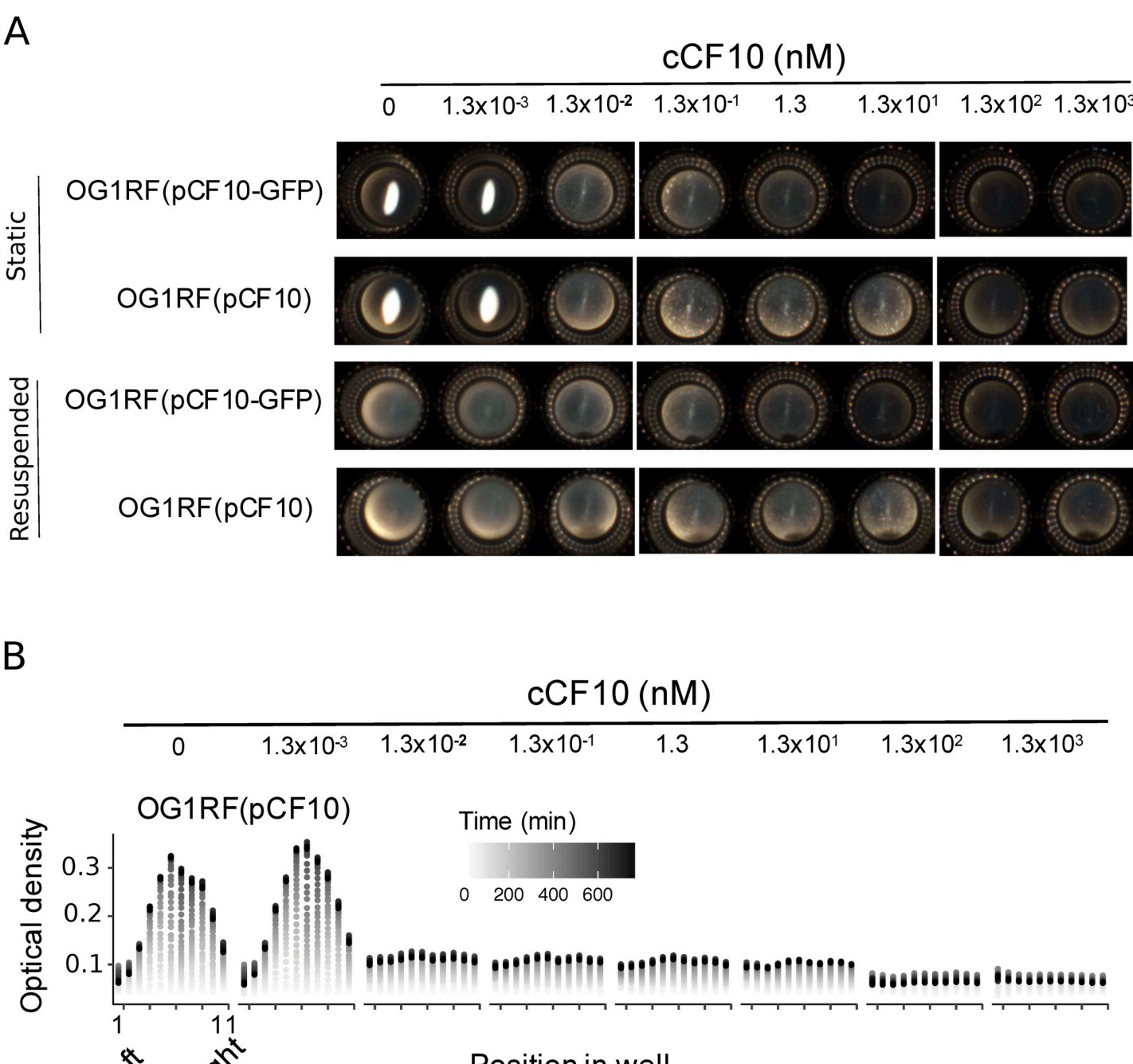

**Fig 3. Response to purified cCF10 is sensitive and costly.** (A) A phenotypic assay for Asc10 activity (see *Adherence/growth assay*) showing the macroscopic appearance of static cultures of OG1RF donors carrying wild-type pCF10 or the pCF10-GFP Asc10 expression reporter [23] after an overnight incubation with varying concentrations of purified cCF10. Unstimulated cells appear as a precipitate in the center of the microwell. Adhered cells appear as a film. (B) Time-resolved microwell horizontal spatial scanning (11 positions) of optical density in OG1RF(pCF10). The underlying numerical data are shown in S4 Data. GFP, green fluorescent protein.

Donor population ($d$) is governed by three dynamics: first, an irreversible second-order mass-action–like process that depicts horizontal plasmid transfer (conjugation), transforming recipients ($r$) to donors with a rate constant ($\lambda_{conj}$), weighed by function $h(r,d)$, which takes values between 0 and 1 and whose form depends on the strategy analyzed; second, a logistic

growth law (i.e., vertical transfer), with maximal rate $\lambda_d$ limited by the carrying capacity $K$ and hindered by the cost of mating activation up to a constant c, as estimated from our experiments (c = 0.2; S5 Fig); and third, an imposed constant dilution rate ($\mu$):

$$d' = \lambda_{conj} \cdot h(r,d) \cdot d \cdot r + \left( \lambda_d \cdot (1 - c \cdot h(r,d)) \cdot \left( 1 - \frac{r+d}{K} \right) - \mu \right) \cdot d \qquad (1)$$

Likewise, recipients grow logistically with a maximal rate $\lambda_r$, yet they are removed from the pool by dilution and upon conjugation:

$$r\prime = -\lambda_{conj} \cdot h(r,d) \cdot d \cdot r + \left( \lambda_r \cdot \left( 1 - \frac{r+d}{K} \right) - \mu \right) \cdot r \qquad (2)$$

We assume that the presence of the plasmid burdens donor proliferation, i.e., $\lambda_d < \lambda_R$. Therefore, plasmid transfer decreases the fraction of recipients, and growth competition increases it. Using this approach, we compared four different strategies ($h(r,d)$): a strategy with constitutive activation (a constant value), a recipient-sensing strategy (a sigmoidal, Hill-type function of the recipient density), a population-sensing strategy (a Hill function of the total density), and a ratio-sensing strategy (a Hill function of the population ratio) (see *Mathematical modeling*).

In principle, such strategies may result in recipient or donor takeover or, alternatively, reach a nontrivial coexistence steady state (see example in S7 Fig). We find that ratio sensing outperforms all other strategies, maximizing the donor population size at any carrying capacity ($K$) value (Fig 4A). Furthermore, the estimated response sensitivity to recipient-to-donor ratio ($\theta$, see *Mathematical modeling*; Fig 2C), shows that the ratio-sensing strategy achieves the highest predicted plasmid fitness relative to other strategies with their own varying sensitivities (S8 Fig). Interestingly, ratio sensing additionally prevails as the only strategy allowing coexistence to be maintained across a wide range of carrying capacity ($K$) values (Fig 4B, S8 Fig).

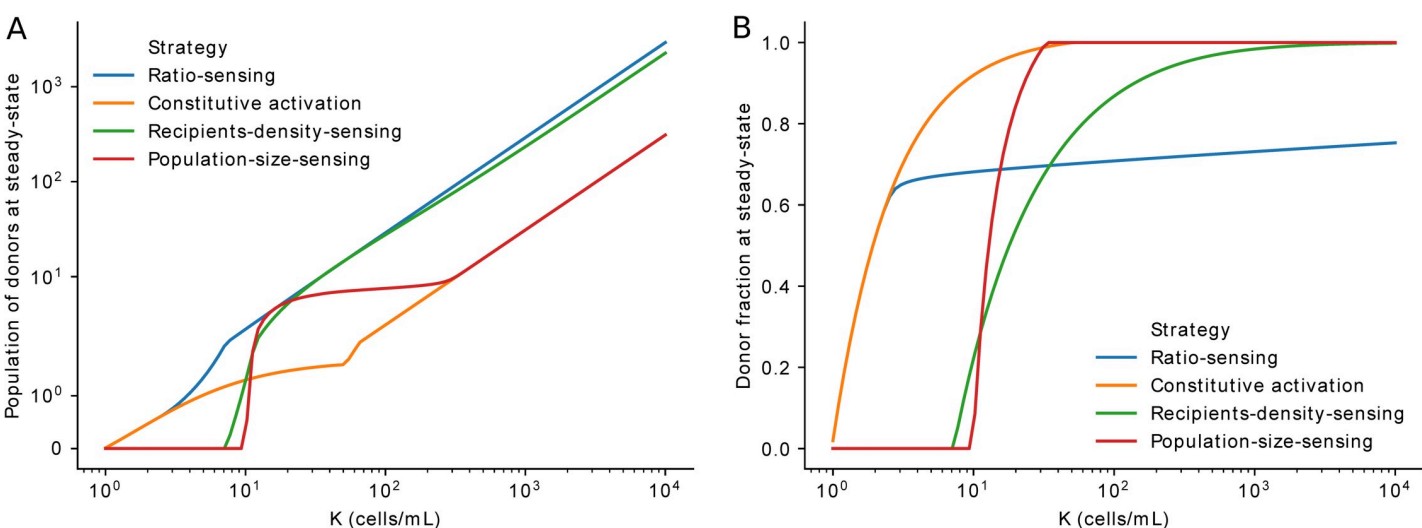

**Fig 4. Mathematical model of mating population dynamics.** (A, B) Steady-state donor density (A) and donor fraction (B) for different sensing strategies as a function of the carrying capacity ($K$). The underlying numerical data are shown in S5 Data. Python code with model and simulations can be found in https://gitlab.inria.fr/InBio/Public/efaecalis-ratio.

## Discussion

Here, we demonstrated that *E. faecalis* pCF10 plasmid donors perform ratiometric sensing, largely agnostic to changes in total density at the tested ranges of population parameters. In very diluted cultures below the tested density range, we expect ratiometric sensing to be unlikely because pheromone concentrations fall below the absolute threshold for mating activation. Additionally, although ratios could be recipient biased at these densities, the overall encounter probability is rather low, making ratiometric sensing unnecessary.

Ratiometric sensing was recently described for other binary populations, such as those composed of signal producers and signal "cheaters" in the PhrA-RapA-Spo0F signaling pathway in *Bacillus subtilis* [26] and yeast mating types [24]. In both cases, ratiometric sensing is achieved by cell density–dependent reductions in extracellular signal availability. In *B. subtilis*, the ratio of signal producers to nonproducers is sensed as a result of the role of population-wide signal internalization or "pumping in" (through the oligopeptide permease [Opp]) in depleting the available extracellular signal. The *E. faecalis* system uses molecular components homologous to those used in *B. subtilis* (Fig 1C). However, two important differences between these systems suggest that the dominant mechanism for ratio sensing in *E. faecalis* relies on signaling rather than pumping alone. First, in the absence of functional iCF10, mating activation is constitutively high in *E. faecalis* [19], making pheromone sensing impossible. Second, signal uptake is donor specific, which can reduce the capacity for ratio sensing by poor global uptake [26], especially in recipient-biased populations. Ratio sensing through iCF10 could have become necessary as PrgZ-dependent, cCF10-specific uptake evolved, both sensitizing the pathway and making recipients comparatively weaker pumps. Despite this, pumping itself can indeed affect extracellular concentrations of both signals. We speculate that even though pumping can modify extracellular signal availability, its effects remain equal for both pheromones, leaving the pheromone ratio unaltered. Nevertheless, the exact relative contributions of signal depletion and iCF10 signaling in shaping the ratiometric response function remain to be uncovered.

Why would such ratiometric sensing evolve? Overall, our modeling results shows that ratiometric sensing is superior to competing strategies because it maximizes plasmid fitness. Our simulations show that ratio sensing maximizes donor concentrations at steady state by letting faster-growing recipients reach a certain fraction of the population before transferring the plasmid (S7 Fig). Therefore, maintaining recipients as a substrate for spread seems more efficient than simply taking over and relying on a comparatively slower vertical spread, maximizing plasmid proliferation in the absence of antibiotics.

Overall, our results suggest that conjugation could in principle be altered externally as a possible therapeutic to artificially drive antibiotic-resistant cells to extinction.

## Materials and methods

### Bacterial strains and culture conditions

Bacterial strains used in this study are listed in Table 1. For gene expression quantification in coincubation (mixed-population) experiments, the OG1RF(pCF10-GFP) reporter strain was used as the donor [23], and OG1RFSSp was used as the recipient cell. For mating experiments, strain OG1RF-GFP (expressing GFP constitutively, [27]) was used as recipient, and the wild-type OG1RF-pCF10 was used as donor. In general, overnight cultures were grown in brain heart infusion (BHI) media and appropriate antibiotics, whereas MM9YEG [28] semisynthetic media was used for day cultures, inductions, and coincubation experiments. The characterization of OG1RF(pCF10-GFP) as a useful reporter for coincubation experiments (see *Coincubation experiments* below) was done by comparing it to a 2-fold-dimmer wild-type GFP reporter

**Table 1. Strains used in this study.**

| *E. faecalis* strain | Description | Reference |
|---|---|---|
| OG1RF(pCF10) | Wild-type OG1RF carrying wild-type pCF10 plasmid. | [29] |
| OG1RF (pCF10-GFP) | pCF10 with a, RBS*prgB*-GFP construct located between prgB and prgC. The cassette disrupts the small *prgU* ORF[1]. Alternative name: OG1RF(pCF10-LC1). | [23] and sequencing results |
| OG1SSp | Wild-type recipient. | [30] |
| OG1RF-GFP | Wild-type expressing GFP constitutively. Alternative name is SD234. | [27] |
| OG1RF(pCF10-*prgC*-GFP) | Wild-type pCF10 GFP construct located downstream of *prgC*. | Gift from Gary Dunny and Wei-Shou Hu |

[1]This ORF was unknown at the time of publication of the referred work.

Abbreviations: GFP, green fluorescent protein; ORF, open reading frame; prg, pheromone responding gene; RBS, ribosome binding site

strain OG1RF(pCF10-*prgC*-GFP) carrying a GFP downstream of *prgC* (a kind gift from Gary Dunny and Wei-Shou Hu).

## Pheromone stimulation

The cCF10 (synthesized by Biocat) pheromone was dissolved in 100% DMSO. Serial dilutions of cCF10 were prepared in DMSO and then added to 96- or 24-well plates (costar) containing the test cultures at an equal optical density of approximately 0.1 using an adjustable-spacer multichannel pipet or a multichannel pipet. Stimulations have a constant final concentration of 0.1% DMSO (which does not affect growth) and 1% bovine seroalbumin (BSA, Sigma) to block peptide adhesion to surfaces. Reported pheromone concentrations are initial, as donor uptake modifies the extracellular concentration with time.

## Coincubation experiments

After day culture growth, donor and receiver strains were mixed at different ratios and immediately serially diluted in 24-well plates, sealed (Breath-easy seals, Diversity Biotech), and shaken at 37˚C (Infors orbital incubator). Strain OG1RF(pCF10-GFP) harbors a GFP construct located downstream of *prgB* ([23], Fig 1C), which is inserted within the *prgU* gene, as determined by sequencing. This reporter shows a high output signal, which is crucial to distinguish populations from autofluorescent recipients (S1 and S2 Figs). Also, the strain does not mate efficiently, further allowing us to minimize the effects of transconjugants that may alter the recipient-to-donor ratio tested. The strain nevertheless maintains normal sexual aggregation and, importantly, remains sensitive in the same physiological range of pheromone stimulation as the wild type (S2 Fig). To calculate the fraction of stimulated donors, we counted the stimulated donors (S1 Fig, gate R1 plus gate R3) and divided it by the total number of donors, which is the total number of cells in the sample: [gate R1 + gate R2 + 2×R3 (each event in R3 is a cell pair)] multiplied by the donor fraction, which is a known variable ab initio.

## Flow cytometry

Before quantification, homogenization of self-aggregates was done mechanically by pipetting up and down approximately 50 times using a multichannel pipet. Cultures were immediately diluted 4-fold, and sampling was performed quickly in a flow cytometer. Flow cytometry was performed in a Gallios instrument (Beckman Coulter) or a Fortessa HT (Becton Dickinson) with sample

sizes of 10,000 to 1,000,000 cells. The fluorescence signal from GFP was acquired using a 488-nm laser for excitation and a 525/40 (Gallios) or a 530/30 filter (Fortessa) for emission.

### Adherence/growth assay

Cell densities in the absence of aggregates were reconstructed from spatially resolved OD600 measurements done on glass-bottom microwell plates (Corning) in a microplate reader (Tecan-Infinity, equipped with a monochromator) with an incubation temperature of 37°C, no shaking, and the "lid on" mode activated. The lid of the plate was kept on and sealed with punctured parafilm to avoid evaporation. Eleven positions within each microwell were acquired at regular intervals. After the experiment finalized, images were acquired with a SCAN 1200 colony counter. To avoid uneven illumination, only three microwells centered per picture were acquired.

### Aggregation and turbidity assay

Aggregation levels were determined in 24-well plates with shaking at 150 RPM at 37°C containing BSA 1% in MM9YEG [28]. Quantification was done by measuring the optical density decrease in time (S4A Fig). Aliquots were sampled near the liquid surface, touching the well wall with the pipet tip. The total turbidity loss (S4B Fig) was determined by sampling the culture at the last time point in S4A Fig after strong resuspension and an immediate dilution in MM9YEG.

### Mating experiment

Mating assays were performed by coincubating the wild-type OG1RF(pCF10) with constitutively fluorescent strain OG1RF-GFP [28] for 2 or 3 hours under optimal conditions for aggregate formation, i.e., in 24-well plates (Costar) under orbital shaking at 150 RPM in a final volume of 1 ml in the presence of 1% BSA. Homogenization of macroscopic sexual aggregates formed during mating reactions was done mechanically by pipetting up and down approximately 50 times using an adjustable-spacer multichannel pipet (Rainin) set to half of the total reaction volume (0.5 ml) and immediately diluting the samples to the appropriate dilution. Dilutions were plated in BHI agar medium. When colonies achieved an appropriate size, a velvet colony replicator was used to stamp the colonies on BHI containing tetracycline (10 μg/ml) to select for donors and transconjugants (and, by plate comparison, identify recipients), which we further distinguished from each other by detecting GFP-positive colonies (transconjugants) in a transilluminator equipped with a camera (Biorad Chemidoc). The mating efficiency was calculated by dividing the number of transconjugants by the number of donors.

### Mathematical modeling

The different donor ($d$) strategies to sense recipients ($r$) were modeled with the Hill-type function $h(r,d)$ varying between 0 and 1 as follows. For the ratiometric-sensing strategy $h = \frac{r^\eta}{r^\eta + (\theta \cdot d)^\eta}$, where $\theta$ is the recipient-to-donor ratio producing half-maximal activation and $\eta$ is the Hill coefficient; for the mate-sensing strategy, $h = \frac{r^\eta}{r^\eta + r_{50}^\eta}$, where $r_{50}$ is the recipient concentration producing half-maximal activation; for the total density-sensing strategy, $h = \frac{P^\eta}{P^\eta + P_{50}^\eta}$, where $P_{50}$ is the recipient concentration producing half-maximal activation; and for constant activation, $h = a$, where $a$ is a constant. Fixed parameter values for simulations (Fig 4, S8 Fig) are shown in Table 2. Coding was done on Python 3.7 with packages sympy 1.4, matplotlib 3.1.1, and scipy 1.3.1. The "BDF" method of the "solve_ivp" function was used to integrate ODEs.

**Table 2. Fixed model parameter values.**

| Parameter | Value | Origin |
|---|---|---|
| $\lambda_{\mathrm{conj}}$ | 0.012 ml·cell$^{-1}$·min$^{-1}$ | From [19] |
| $\theta$ | 0.5 | Estimated from Fig 2C |
| $\eta$ | 4.3 | Estimated from Fig 2C |
| $\lambda_D$ | 0.01548 min$^{-1}$ | From [19] |
| $\lambda_R$ | 0.0201 min$^{-1}$ | From [19] |
| $\mu$ | 0.012 min$^{-1}$ | Arbitrary |
| $c$ | 0.2 | Estimated from S5 Fig |

## Supporting information

**S1 Fig. Identification of activated donors in mixed-population experiments.** (A, B) Scatter plot of pure donor OG1RF(pCF10-GFP) (A) and recipient OG1SSp (B) cell populations. (C) Example of a coincubation reaction of donors and recipients at a recipient-biased ratio. The mean single-cell intensity of donors was obtained from population R1. Population R2 corresponds to a mixture of autofluorescent donors and nonactivated recipients. Population R3 corresponds to donor–recipient pairs (and higher-order aggregates, i.e., events in R3 with higher fluorescence values than events in R1 ["tail"] were not considered in the analysis). To estimate the fraction of activated donors (Fig 2C), we considered the three populations and the known experimentally determined initial ratio in the calculation (see *Coincubation experiments*). (D) Histogram with GFP intensities from the data in (C). The underlying numerical data are shown in S6, S7 and S8 Data. GFP, green fluorescent protein.
(TIF)

**S2 Fig. Fluorescent Reporter comparison.** Comparison of (OG1RF(pCF10-GFP)) (Δ*prgU*, increased output) and wild-type (OG1RF(pCF10-*prgC*-GFP)) cCF10 dose-responses. Error bars are SEM (*n* = 4). The underlying numerical data are shown in S9 Data. GFP, green fluorescent protein. *prg*, pheromone responding gene.
(TIF)

**S3 Fig. Self-aggregate formation.** Orbital shaking–dependent macroscopic self-aggregate formation in OG1RF(pCF10) as a function of cCF10 concentration. The physiological (heterophilic) range of pheromone concentration required to induce detectable Asc10 expression lies below 1 nM cCF10.
(TIF)

**S4 Fig. Pheromone-dependent WT growth under shaking.** (A, B) WT (OG1RF(pCF10)) versus OG1RF(pCF10-prgC-GFP) reporter comparison in an aggregate formation assay (A) and postaggregate dispersion cell density measurement (B) (see *Aggregation and turbidity assay*). Note the increase in turbidity at 1 hour in (A), the time at which aggregation brings it back to the baseline. The underlying numerical data are shown in S10 Data. GFP, green fluorescent protein; prg, pheromone responding gene; WT, wild-type.
(TIF)

**S5 Fig. Pheromone-dependent wild-type growth in static cultures.** Growth reduction in the physiological (sub-nanomolar) response range in growth/adhesion assays (see *Adherence/growth assay*). Error bars are SD of *n* = 3 independent experiments (colors). The underlying numerical data are shown in S11 Data.
(TIF)

**S6 Fig. PrgU suppresses excessive fitness costs associated with unproductive activation of conjugation.** (A, B) Growth/adhesion assay showing mean growth (per well) of pCF10 (A, from Fig 3C) and pCF10-GFP ($\Delta prgU$) (B). The underlying numerical data are shown in S12 Data. GFP, green fluorescent protein; Prg, pheromone responding gene.
(TIF)

**S7 Fig. Example of simulated growth dynamics.** Representative example of a simulation showing the dynamics of (in this case) ratio-sensing strategy with different starting fraction of recipients and total population sizes. The underlying numerical data are shown in S13 Data.
(TIF)

**S8 Fig. Ratio sensing maximizes donor fitness (top) and helps coexistence (bottom).** Strategy comparison showing extended simulations varying relevant parameter values for each strategy: activation level (*a*) for the constitutive activation strategy and sensitivity (half-maximal activation; $P_{50}$, $\theta$, and $r_{50}$; see *Mathematical modeling*) for the rest. Each value is normalized by the maximum value observed within each *K*. The underlying numerical data are shown in S14 Data.
(TIF)

**S1 Data. CSV file containing the underlying numerical data for Fig 2A and 2B.**
(CSV)

**S2 Data. CSV file containing the underlying numerical data for Fig 2C.**
(CSV)

**S3 Data. CSV file containing the underlying numerical data for Fig 2D.**
(CSV)

**S4 Data. CSV file containing the underlying numerical data for Fig 3B.**
(CSV)

**S5 Data. CSV file containing the underlying numerical data for Fig 4A and 4B.**
(CSV)

**S6 Data. CSV file the underlying single-cell data for S1A Fig.**
(CSV)

**S7 Data. CSV file the underlying single-cell data for S1B Fig.**
(CSV)

**S8 Data. CSV file the underlying single-cell data for S1C and S1D Fig.**
(CSV)

**S9 Data. CSV file containing the underlying numerical data for S2 Fig.**
(CSV)

**S10 Data. CSV file containing the underlying numerical data for S4A and S4B Fig.**
(CSV)

**S11 Data. CSV file containing the underlying numerical data for S5 Fig.**
(CSV)

**S12 Data. CSV file the underlying numerical data for S6A and S6B Fig.**
(CSV)

**S13 Data. CSV file containing numerical data for S7 Fig.** Time traces 1–8 correspond to lowest to highest initial densities in the legend of S7 Fig.
(CSV)

**S14 Data. CSV file containing the underlying numerical data for S8 Fig.**
(CSV)

## Acknowledgments

We thank Gary Dunny, Wei-Shou Hu, Ivan Matic, Dawn Manias, Rebecca Breue, Laura Cook, Christopher Kristich, Barbara Murray, Danielle Garsin, and Lynn Hancock for valuable advice and/or strains.

## Author Contributions

**Conceptualization:** Alvaro Banderas, Ariel B. Lindner.

**Formal analysis:** Alvaro Banderas, Arthur Carcano.

**Funding acquisition:** Alvaro Banderas, Ariel B. Lindner.

**Investigation:** Alvaro Banderas, Arthur Carcano, Elisa Sia, Shuang Li.

**Methodology:** Alvaro Banderas, Arthur Carcano.

**Project administration:** Alvaro Banderas.

**Resources:** Ariel B. Lindner.

**Software:** Arthur Carcano.

**Supervision:** Alvaro Banderas, Ariel B. Lindner.

**Validation:** Alvaro Banderas.

**Writing – original draft:** Alvaro Banderas.

**Writing – review & editing:** Alvaro Banderas, Arthur Carcano, Ariel B. Lindner.

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
