## [Editor Report · Decision Letter 0]

21 Feb 2020

Dear Dr Banderas, 

Thank you for submitting your manuscript entitled "Ratiometric quorum sensing in Enterococcus faecalis conjugation" for consideration as a Research Article by PLOS Biology.

Your manuscript has now been evaluated by the PLOS Biology editorial staff, as well as by an academic editor with relevant expertise, and I'm writing to let you know that we would like to send your submission out for external peer review. IMPORTANT: We note that you have submitted this as a full Research Article, but we think that it would be better reviewed as a Short Report. No re-formatting is required, as your paper is already quite concise, but please change the article type to "Short Reports" when you upload your additional metadata (see next paragraph).

Please re-submit your manuscript within two working days, i.e. by Feb 25 2020 11:59PM.

Kind regards,

Roli Roberts

Senior Editor

PLOS Biology

---

## [Decision Letter · Decision Letter 1]

19 Mar 2020

Dear Dr Banderas,

Thank you very much for submitting your manuscript "Ratiometric quorum sensing in Enterococcus faecalis conjugation" for consideration as a Short Report at PLOS Biology. Your manuscript has been evaluated by the PLOS Biology editors, an Academic Editor with relevant expertise, and by three independent reviewers.

You'll see that all three of the reviewers are broadly positive about your study. However, reviewers #2 and #3 raise a significant number of concerns, mostly involving your mathematical model and its relationship to the rest of the manuscript. These issues will need to be addressed before we can consider your manuscript further.

In light of the reviews (below), we will not be able to accept the current version of the manuscript, but we would welcome re-submission of a much-revised version that takes into account the reviewers' comments. We cannot make any decision about publication until we have seen the revised manuscript and your response to the reviewers' comments. Your revised manuscript is also likely to be sent for further evaluation by the reviewers.

We expect to receive your revised manuscript within 2 months; however, we're very aware of the current global problems caused by Covid-19, and are prepared to be very flexible about this time-frame. Please email us (plosbiology@plos.org) if you have any questions or concerns, or would like to request an extension. At this stage, your manuscript remains formally under active consideration at our journal; please notify us by email if you do not intend to submit a revision so that we may end consideration of the manuscript at PLOS Biology.

**IMPORTANT - SUBMITTING YOUR REVISION**

*Re-submission Checklist*

*Published Peer Review*

*PLOS Data Policy*

*Blot and Gel Data Policy*

Sincerely,

Roli Roberts

Senior Editor

PLOS Biology

REVIEWERS' COMMENTS:

Reviewer #1:

This manuscript focuses on defining the parameters and mechanism that allow plasmid-bearing antibiotic resistant cells to avoid being outcompeted in the absence of antibiotics. The authors evaluate several potential sensing mechanisms and conclude that ratiometric sensing of donor:recipient cells drives sexual behavior in bacteria. This is a well-executed study that report provocative results that should be of high interest to researchers studying sociomicrobiology and evolution. I also believe that the conclusions of this study are well-supported by the experimental data. I therefore recommend publication in PLOS Biology.

Reviewer #2:

Banderas et al. present an interesting set of data on the function of the E. faecalis extracellular signaling system in the regulation of conjugation. A key finding is that the response changes a function of the composition of the population rather than by absolute cell density. This result is very convincingly demonstrated in Fig. 2. The authors also provide evidence that there is a trade-off between investing in horizontal versus vertical plasmid transfer as the induction of the pheromone pathway reduces the growth rate (Fig. 3). Finally, the authors present a model that could explain how ratio-sensing contributes to a stable equilibrium of donor-recipient ratio in the population based on a mathematical model (Fig. 4).

I found the MS interesting and thought provoking and generally feel that it could qualify as a Short Communication. However, I have a few remarks which should be considered.

Major comments

1. Ratio-sensing and signaling architecture: The authors motivate their ratio-sensing hypothesis by highlighting specific features of regulatory architecture of the signaling system. In Figure 1 the authors propose that the ratio-sensing ability requires a) two signaling systems and they assume b) that the extracellular concentrations of both signals are proportional to the donor and recipient concentrations, respectively. These assumptions are not tested in the MS and they may not hold.

It seems important to point out that the investigated signaling system functions by means of a peptide-based export-import circuit (a wide-spread signaling architecture found in many G+ bacteria, Neiditch et al., Annu. Rev. Genet. 2017. 51:311-33, https://doi.org/10.1146/annurev-genet-120116-023507 ). The authors do not mention/consider this aspect. However, this could be important to explain the phenomena observed by the authors. In a recent publication it was shown that export-import (or „pump-probe" signaling systems) extracellular concentrations do not necessarily increase with increasing population density. Moreover, such systems are capable of ratio-sensing in mixed populations of producer and non-producer cells, when signals are taken up very efficiently compared to overall signal production in the population (Babel et al., Nat. Commun. 11, 2020, https://doi.org/10.1038/s41467-020-14840-w)

Hence, for ratio-sensing in E- faecalis conjugation, the second signaling system (assumption a) might in fact be dispensible and assumption b is by no means trivial and not necessarily to be expected. This should be considered/discussed. Please either provide additional data to support their specific model or more carefully introduce/motivate/discuss the ratio-sensing hypothesis adopting a broader mechanistic perspective of the overall signaling architecture.

2. Stimulation experiments to support tradeoff model: Again, given the network architecture, in the stimulation experiments performed in Fig. 3 it is not obvious that the extracellular concentration is indeed the relevant biophysical quantity as the signals will be imported by the cells and activate an intracellular receptor. It might be the available dose of signaling molecules not the extracellular concentrations of signaling molecules present at the start of the experiment, which matters. This point will likely not affect the conclusions of the experiment, since presumably cell densities were initially all the same in all experiments. Thus variation in concentrations are equivalent to variation in dose. Nevertheless, this point should at least be mentioned or alternatively clarified experimentally.

3. Role of prgU/mathematical model: Here I got confused how the tradeoff got linked to the model.

Looks like the authors caveat against the the tradeoff which they just demonstrated and then come up with an alternative hypothesis for the „function(?)" of ratio-sensing. Please clarify. Also consider to replace the title „Mathematical model for conjugation dynamics" by a statement that states the findings derived from the model. I was also confused by the intended message, is the coexistence a „function" of ratio-sensing or a „consequence"? The discussion then brings up the antibiotics for this part. The presentation of this part is sub-optimal and needs improvement for clarity.

Minor comments

4. Check for typos in the text and the figures! e.g. Fig 1a. plamid/plasmid, simulation/stimulation

5. "monitoring the expression of a GFP reporter which is driven by a copy of

prgB's ribosome binding site (RBS) further downstream in the transcript". Maybe reword to facilitate easier understanding.

6. Fig. 2A/B. Please explain the color bar in the caption.

7. Mathematical model: check consistency of notation for all parameters.

8. „If 50 is however of an order of magnitude between those of (1 −/) and (1 -/), then the system behavior is not analysed here." 

Please explain a bit more.

Reviewer #3:

In this manuscript, Banderas et al. study the regulation of plasmid conjugation in the pathogenic bacterium Enterococcus faecalis. They show experimentally that conjugation is induced when the bacterial population is composed mostly of recipient (plasmid-free) cells, rather than when the overall population density is sufficiently high. Further experiments show that activation of conjugation is costly. The authors therefore argue that ratiometric control of conjugation mitigates this cost and ensures that donors only bear the cost of conjugation when recipients abund and the benefit of conjugation is high. Finally, the authors construct a mathematical model to study how the interplay between the costs and benefits of plasmid carriage affects the prevalence of plasmid-bearing cells in bacterial populations. 

Overall, I find this work to be of high quality, novel and of broad interest. Ratiometric control of conjugation makes intuitive sense, and can significantly affect the dynamics of plasmid-born traits, such as pathogenicity and antibiotic resistance. Yet, I am not familiar with previous works discussing this mode of regulation and its implications. I do have major comments regarding the model as well as more minor ones, as detailed below. 

Major comments

-------------------

As formulated, the model is inadequate for its declared purpose - comparing conjugation regulation strategies. This is because the model only accounts for the constant cost of plasmid carriage and does not consider the cost of activating conjugation - a cost demonstrated in Fig. 3. Therefore, in this model activation of conjugation at any population density or composition can only be beneficial for plasmid spread. Indeed that is the case in the results shown in Fig. 4B, where ratio-sensing results in the lowest prevalence of donors. 

Moreover, I found the modeling section to be misleading, and at odds with the rest of the paper. The conclusion of this section is that ratio-sensing is "the only strategy allowing a robust co-existence of the two populations [donors and recipients]". Since in previous sections the authors argue that ratio-sensing may be optimal for the plasmid in mitigating the demonstrated cost of conjugation, I initially took that conclusion to mean that ratio sensing prevents the extinction of donor, thus allowing co-existence of recipient and donors. However, the model results in fact show the exact opposite - ratio sensing prevents the extinction of recipients. 

Since in this model ratio sensing is not beneficial to the plasmid which encodes the regulatory machinery implementing this regulation it is also not clear why such regulation would evolve. In the discussion, the authors argue that maintaining coexistence between carriers and plasmid-free cells may be beneficial to the population as a whole under intermittent antibiotic exposure. This is an interesting idea, and ratio sensing may be an evolutionary stable state under specific conditions. However, a simpler, and likely more robust mechanism for the evolution of ratio sensing is that it is directly beneficial to the plasmids that implement it. 

Therefore, the model needs to be revised to include a cost for conjugation. I realize this entails some arbitrary modeling decisions and the addition of parameters to the model. However, I believe that any reasonable choice can dramatically change the behavior of the model, and make it suitable to address the question of when is ratio sensing favorable to the donors. Since this manuscript is being considered as a Short Report, a simple "proof-of-concept" model showing the benefit of ratio sensing for plasmid donors would suffice. Such a model would likely motivate subsequent more comprehensive dedicated modeling efforts. 

Additional comments

*The experiments demonstrating the cost of conjugation do not rule out the possibility of cCF10 toxicity. The claim would be significantly strengthened by additional experiments showing that plasmid-free recipient strains are unaffected by exposure to the same concentrations of cCF10.

*I find it unlikely that conjugation regulation is insensitive to population density even at very low population densities, as stated in the text and implied in Fig. 1B. At low population densities, it would be unlikely for a donor to encounter a recipient even at a high R:D ratio. The current experiments demonstrating ratiometric regulation are all done in very high population densities (OD 0.1-1), therefore it is still unknown how conjugation is regulated at low population densities. I am not advocating that the authors conduct further experiments, since the novelty is in the fact that ratiometric regulation occurs at all. But, this caveat should be stated and discussed. 

*More information is required regarding the experiments shown in Fig. 3B. Data from how many replicates is shown? Or is it a single replicate per condition? 

*Additionally, the histograms to the right have some white dots in them. Is that an issue with the rendering?

*I would replace Fig. 4A, with a heatmap showing the steady-state donor fraction as a function of initial total population size and donor fraction. 

*While arbitrary, the choice of nM units for the total population size 'K' in panel B is unusual and confusing. Suggest replacing it with cells/ml.

---

## [Decision Letter · Decision Letter 2]

16 Jun 2020

Dear Dr Banderas,

Thank you for submitting your revised Short Report entitled "Ratiometric quorum sensing in Enterococcus faecalis conjugation" for publication in PLOS Biology. I have now obtained advice from two of the original reviewers and have discussed their comments with the Academic Editor. 

We're delighted to let you know that we're now editorially satisfied with your manuscript. However before we can formally accept your paper and consider it "in press", we also need to ensure that your article conforms to our guidelines. A member of our team will be in touch shortly with a set of requests. As we can't proceed until these requirements are met, your swift response will help prevent delays to publication. Please also make sure to address the data and other policy-related requests noted at the end of this email.

IMPORTANT:

a) Please can you make your Title more accessible? Given the implications your results have for plasmid maintenance and transmission of antimicrobial resistance, etc., your current title may not attract readers that would in principle be interested in the conclusions.

b) Please attend to my Data Policy requests further down.

c) Please supply any custom code needed to reproduce your results, either as supplementary files, or by depositing on e.g. GitHub.

*Copyediting*

*Published Peer Review History*

*Early Version*

*Submitting Your Revision*

Sincerely,

Roli Roberts

Senior Editor

PLOS Biology

DATA POLICY:

Regardless of the method selected, please ensure that you provide the individual numerical values that underlie the summary data displayed in the following figure panels as they are essential for readers to assess your analysis and to reproduce it: Figs 2ABCD, 3B, 4, S1 (FACS), S2, S4AB, S5, S6AB, S7, S8. NOTE: the numerical data provided should include all replicates AND the way in which the plotted mean and errors were derived (it should not present only the mean/average values).

REVIEWERS' COMMENTS:

Reviewer #2:

The manuscript has improved in clarity and my concerns have been addressed by the authors.

Reviewer #3:

I thank the authors for their detailed response. All of my concerns have been addressed, and I am happy to recommend the this manuscript for publication.

---

## [Editor Report · Decision Letter 3]

20 Jul 2020

Dear Dr Banderas,

On behalf of my colleagues and the Academic Editor, Nathalie Balaban, I am pleased to inform you that we will be delighted to publish your Short Reports in PLOS Biology. 

Early Version

PRESS 

Kind regards,

Vita Usova

Publication Assistant, 

PLOS Biology

on behalf of

Roland Roberts,

Senior Editor

PLOS Biology